# The Roles of Periodontal Bacteria in Atherosclerosis

**DOI:** 10.3390/ijms241612861

**Published:** 2023-08-16

**Authors:** Xiaofei Huang, Mengru Xie, Xiaofeng Lu, Feng Mei, Wencheng Song, Yang Liu, Lili Chen

**Affiliations:** 1Department of Stomatology, Union Hospital, Tongji Medical College, Huazhong University of Science and Technology, Wuhan 430022, China; fff_1118@hust.edu.cn (X.H.); hbxiemengru@hust.edu.cn (M.X.); lxf_yuki@hust.edu.cn (X.L.); hustmeifeng@hust.edu.cn (F.M.); d202280813@hust.edu.cn (W.S.); 2School of Stomatology, Tongji Medical College, Huazhong University of Science and Technology, Wuhan 430030, China; 3Hubei Province Key Laboratory of Oral and Maxillofacial Development and Regeneration, Wuhan 430022, China

**Keywords:** atherosclerosis, CVD, periodontal pathogens, *Porphyromonas gingivalis*, *Aggregatibacter actinomycetemcomitans*, *Fusobacterium nucleatum*, plaque, instability

## Abstract

Atherosclerosis (AS) is an inflammatory vascular disease that constitutes a major underlying cause of cardiovascular diseases (CVD) and stroke. Infection is a contributing risk factor for AS. Epidemiological evidence has implicated individuals afflicted by periodontitis displaying an increased susceptibility to AS and CVD. This review concisely outlines several prevalent periodontal pathogens identified within atherosclerotic plaques, including *Porphyromonas gingivalis*, *Aggregatibacter actinomycetemcomitans*, and *Fusobacterium nucleatum*. We review the existing epidemiological evidence elucidating the association between these pathogens and AS-related diseases, and the diverse mechanisms for which these pathogens may engage in AS, such as endothelial barrier disruption, immune system activation, facilitation of monocyte adhesion and aggregation, and promotion of foam cell formation, all of which contribute to the progression and destabilization of atherosclerotic plaques. Notably, the intricate interplay among bacteria underscores the complex impact of periodontitis on AS. In conclusion, advancing our understanding of the relationship between periodontal pathogens and AS will undoubtedly offer invaluable insights and potential therapeutic avenues for the prevention and management of AS.

## 1. Introduction

Atherosclerotic cardiovascular disease (CVD) is a major public health problem of all humankind. It is the primary contributor to death and disability and accounts for 1/3 of the deaths in the world [1,2]. Atherosclerosis (AS) is one of the most common causes of CVD. Stenosis, obstruction or rupture of blood vessels can lead to ischemic CVDs such as myocardial infarction, stroke, and limb ischemia [3]. AS is considered to be a chronic inflammatory disease of the arterial wall caused by a variety of stimulating factors, characterized by the formation, progression, and instability of atherosclerotic plaques. It often involves medium and large arteries. The development of AS is a long-term and slow accumulation process. It usually begins with the injury of the vascular endothelial barrier and is followed by cholesterol-rich lipoprotein accumulating subcutaneously. Vascular smooth muscle cells (VSMCs) migrate from vascular media to subendothelium, proliferate and synthesize extracellular matrix (ECM), resulting in intimal thickening, which is called diffuse intimal thickening (DIT). Subsequently, the resident VSMCs and blood monocyte-derived macrophages recruited in the subendothelial space uncontrolled uptake modified lipoproteins by scavenger receptors, transforming into lipid-rich cells called “foam cells” and leading to the formation and enlargement of AS plaques. With the death of cells and the disfunction of efferocytosis, the arterial plaque gradually becomes unstable, exhibits necrosis and calcification, or even ruptures and detaches to form a thrombus [4].

Traditional risk factors for AS include lifestyle factors, primarily smoking, dyslipidemia, hypertension, and altered glucose metabolism [5]. Studies in recent decades have revealed that infection plays an important role in AS. Beginning with Fabricant and colleagues, who induced AS in chickens by Marek’s disease virus infection and prevented atherosclerotic changes by vaccination [6], microbial infections such as herpes simplex virus [7], *Chlamydia pneumoniae* [8], *Porphyromonas gingivalis* (*Pg*) [9], *Helicobacter pylori* [10], influenza A virus [11], hepatitis C virus [12], cytomegalovirus [13], and HIV [14] have all been identified as risk factors for AS.

As one of the four major human bacterial reservoirs, more than 700 bacterial species exist in the oral cavity [15,16]. It is worth noting that these bacteria maintain an ecological balance within a healthy periodontium. However, in the presence of periodontal disease, microbial dysbiosis emerges, leading to a shift from Gram-positive anaerobic bacteria to Gram-negative anaerobic bacteria. Consequently, certain bacteria opportunistically acquire pathogenic capabilities, further exacerbating the pathogenesis of the disease [17,18]. Local or systemic infections of oral origin are prevalent in the human population; for example, periodontitis is the sixth most prevalent disease on a worldwide scale, with a global prevalence of 45–50% [19]. It not only contributes to the destruction of local tissues but is also related to the development of a variety of systemic diseases such as AS. The American Heart Association (AHA) acknowledges the correlation between periodontal disease (PD) and atherosclerotic cardiovascular disease, irrespective of known confounding factors [20]. Epidemiologic evidence shows that the incidence of AS in patients with periodontitis is 1.27 times higher than that in patients without periodontitis [21]. In a Japanese study, the prevalence and severity of overall arterial stiffness, as assessed using the Cardio-Ankle Vascular Index, were 2.12-fold higher in elderly patients with severe periodontitis than in a control population (95% CI = 1.09–4.11) [22]. While current evidence does not elucidate a specific predilection of periodontal pathogens for the occurrence sites of AS, research has indicated a significant correlation between periodontitis and AS in the carotid, coronary, and peripheral arteries. In a study conducted by Robert Berent et al., it was found that the prevalence of periodontal disease was 55.6% in patients with coronary heart disease (CHD), while it was 41.9% in non-CHD individuals (*p* < 0.01), underscoring a noteworthy association between PD and CHD (OR = 1.9; 95% CI = 1.2–3.1) [23]. Periodontitis has been demonstrated to correlate with an elevated risk of CHD (OR = 1.24 (95% CI 1.01–1.51) to 1.34 (95% CI 1.10–1.63)), as well as an increased likelihood of experiencing a myocardial infarction (OR = 1.49; 95%CI = 1.21–1.83) [24]. Interventional therapy with stent implantation stands as the principal approach for vascular reconstruction in CVD. However, some patients still develop in-stent restenosis (ISR), which remains a major challenge in current treatments [25]. The existing research has demonstrated that oral infection serves as an independent risk factor for the occurrence of ISR in patients with acute coronary syndrome (ACS) (OR = 1.202, 95% CI = 1.085–1.333, *p*  <  0.01) [26]. Stages III and IV periodontitis are associated with an elevated incidence of ISR following coronary angioplasty (OR = 5.82) [27]. Furthermore, patients who experience ISR after percutaneous coronary intervention tend to have a higher severity of periodontal disease, among which 64.1% are classified as having Stage Ⅳ periodontitis [28]. Similar to coronary arteries, in carotid atherosclerosis, it is crucial to consider not only the extent of stenosis but also certain distinctive features of vulnerable plaques. These characteristics encompass intraplaque hemorrhage, plaque ulcerations, relative volume of calcification, and overall plaque burden. These factors have been demonstrated to offer distinct supplementary insights into the likelihood of recurrent stroke and in-stent restenosis [29,30]. A study revealed a positive correlation between periodontal pathogen load and carotid intima-media thickness (IMT). The average carotid IMT increased from 0.84 to 0.86 mm with the accumulation of bacterial burden, further escalating to 0.87 mm (*p* = 0.04) [31]. In the population-based Malmo Offspring Study, it was validated that moderate to severe periodontitis is significantly connected with an elevated risk of carotid plaque (OR = 1.76; 95% CI = 1.11–2.78) [32]. Through linear regression analysis, it was estimated that each additional periodontal pocket depth of ≥4 mm is projected to increase the total carotid plaque area by 0.34 mm^2^ [32]. In males, there is a significant correlation between periodontitis with high clinical periodontal disease index and calcified carotid artery atheromas (OR = 1.83; 95% CI = 1.28 to 2.64; *p* < 0.01) [33]. Peripheral arterial disease (PAD) is a prevalent condition affecting the blood vessels of the limbs. It manifests as insufficient blood flow to the limbs due to arterial stenosis, commonly resulting in intermittent claudication and other associated symptoms [34,35]. Among the various etiologies of PAD, AS stands as the primary underlying cause [34]. According to a primary healthcare assessment conducted in the United States, approximately 50% of patients with PAD exhibit comorbidities such as coronary artery disease (CAD) or cerebrovascular disease [36]. In comparison to non-PAD individuals, those with PAD have a significantly elevated risk of developing periodontitis and a greater incidence of tooth loss [37,38]. Furthermore, several studies have indicated periodontitis is significantly linked to PAD [37,39,40,41,42], along with a significant correlation between tooth loss and increased PAD risk [43,44]. An investigation conducted among Asian Indians revealed that individuals with periodontitis had higher peripheral arterial pulse wave velocity and arterial stiffness index, indicating greater vascular narrowing and hardening [45].

Oral bacteria can cause temporary bacteremia during some therapy like periodontal treatment, tooth extraction or during daily oral hygiene practices such as chewing, brushing, and flossing, especially in subjects with existing periodontitis and dental pulp infections. Periodontal pathogens can reach distant organs through the blood. Researchers have detected DNA of periodontitis pathogen from atherosclerotic plaques [46,47], providing direct evidence for the link between periodontitis and AS. The main periodontal pathogens detected in the plaques include *Pg*, *Aggregatibacter actinomycetemcomitans* (*Aa*), *Fusobacterium nucleatum* (*Fn*), *Prevotella intermedia* (*Pi*), *Tannerella forsythia* (*Tf*), *Treponema denticola* (*Td*), and *Campylobacter rectus* (*Cr*). In addition, several benign species associated with dental plaque on the tooth surface were detected in the plaques [48].

Periodontitis has been established as a contributing factor in the regulation of AS, but the specific mechanisms remain unclear. In this review, we present the characteristics and virulence factors of several major periodontal pathogens and provide a summary of the available evidence regarding the link between them and AS in humans and animals, as well as highlighting the roles of these pathogens in different pathological processes of AS, with the hope of providing references for future researches on prevention and treatment of AS.

## 2. *Porphyromonas gingivalis*

*Pg* is a dark, lytic, nonmotile, Gram-negative obligate anaerobes that derive energy from the fermentation of amino acids, which facilitates its survival in the subgingival sulcus and periodontal pockets. *Pg* is a main pathogen of periodontitis, and it forms the “red complex” with *Tf* and *Td*, which is responsible for the severe clinical manifestation of periodontal disease. *Pg* is one of the most common bacteria found in the subgingival biofilm of patients with periodontitis [49,50]. A retrospective study conducted in Germany involving 7804 adults diagnosed with periodontitis reported a detection rate of 68.2% for *Pg* in the biofilm of periodontal pockets [51]. *Pg* has a variety of virulence factors, such as lipopolysaccharide (LPS) on the bacterial outer membrane, which can activate the pathogen-related pattern recognition receptor signaling pathway, cause inflammatory response and the secretion of cytokines. Gingipains are trypsin-like cysteine proteinases generated by *Pg* that can cleave laminin, fibronectin, and collagen, activate complement pathways, and induce dysregulation of coagulation and fibrinolytic pathways. Additionally, *Pg* possesses fimbriae, capsules, lipoteichoic acid, hemagglutinin, outer membrane proteins, and outer membrane vesicles. These components facilitate *Pg* to colonize and invade cells, destruct tissue, as well as escape from immune surveillance, inhibit the host immune response, and prolong its survival time.

### 2.1. The Association between Pg Exposure and AS in Humans and Animals

In recent years, the effect of *Pg* in the progress of AS has been confirmed by a large number of studies. Multiple studies have employed polymerase chain reaction (PCR) to examine bacterial presence in human arterial tissues, and *Pg* is among the predominant bacteria frequently detected in atherosclerotic plaques [46,52,53,54]. Adrian Brun’s group utilized a combined approach of nested PCR and real-time PCR to successfully identify *Pg* with high specificity in highly calcified AS plaques of humans [54]. In patients with AS, the serum concentrations of IgG targeting *Pg* and *Pg*-heat shock protein (HSP) 60 are higher than in healthy individuals [55]. Moreover, *Pg*-HSP60 shares common antigenic epitopes with human T cells and/or B cells, suggesting that *Pg* and its HSP60 may be involved in the immunoregulatory processes of AS [55]. A clinical study conducted in Japan demonstrated that patients with type 2 diabetes (DB) with higher serum concentrations of anti-*Pg* IgG presented a higher degree of stenosis in the carotid artery plaque segment compared to patients with lower IgG concentrations (12.0% ± 2.2% for the former, 5.5% ± 1.4% for the latter, *p* = 0.009) [56]. *Pg* was also detected in saliva, supragingival and subgingival plaque of abdominal aortic aneurysm (AAA) patients and the level of *Pg* and plasma antibodies are correlated with AAA diameter and thrombus volume [57]. A case–control study revealed that *Pg* is an independent risk factor for stroke, and elevated levels of *Pg* antibodies in the serum are related to an increased risk of stroke with a multivariate OR of 1.63 (95% CI = 1.06–2.50) for males and an OR of 2.30 (95% CI = 1.39–3.78) for females [58]. Some studies have successively confirmed that *Pg* infection through oral infection, intragastric administration or intravenous injection, can lead to significant enlargement of atherosclerotic plaques in *ApoE*^-/-^ mice-established AS animal model [59,60,61,62], and may worsen the plaque instability [63,64] as well. The level of serum anti-*Pg* IgG and systemic pro-inflammatory cytokines in mice is significantly elevated followed by *Pg* infection [60,65]. *Pg* can also invade the ischemic myocardium of mice with myocardial infarction and promote cardiac rupture, thereby increasing the mortality rate in mice [66]. Below, we will review the promotion effects of *Pg* on AS and its related mechanisms in the order of AS pathological processes.

### 2.2. Mechanistic Investigations on the Effect of Pg in Different Courses of AS

#### 2.2.1. Endothelial Barrier Disruption

Endothelial dysfunction is an early cardiovascular response to stimuli and is considered an “alarm” of AS. After entrance to the blood, *Pg* adheres to the cell surface of endothelial cells (ECs) via a variety of adhesins (including fimA and HagB) to interact with E-selectin, vascular cell adhesion molecule 1 (VCAM-1), intercellular cell adhesion molecule 1 (ICAM-1), and other molecules on the cell surface [67,68,69]. It can also be internalized into ECs by lipid rafts on the cell membrane. *Pg* also stimulates the expression of adhesion molecules, toll-like receptors (TLRs) signaling pathway factors, and chemokine in ECs [70,71,72], while fimA-deficient mutant loses these capabilities [70,71]. Activation of autophagy is one of the mechanisms by which *Pg* escapes the host’s immune defense. After entering the host cell, *Pg* can stimulate the cell autophagy pathway, and be transported from the phagosome to the autophagosome, and to the late autophagosome. There, *Pg* can prevent its maturity as autolytic enzyme autolysosome, thereby avoiding being degraded. *Pg* can also take advantage of the abundant proteins in autophagic vesicles for division and replication [73]. Through inhibition of the autophagy pathway in ECs with 3-methyladenine or Wortmannin, approximately 78% of internalizing *Pg* eventually localized to vacuoles containing cathepsin L and were degraded [73,74]. *Pg* also employs lysine-specific gingipains (Kgp) to initiate the proteolysis of receptor-interacting protein kinase 1 (RIPK1), RIPK2, and poly (ADP-ribose) polymerase (PARP), thereby actively engaging in tumor necrosis factor (TNF)-mediated cell death and nucleotide-binding oligomerization domain (NOD)-mediated host defense pathways. This intricate process facilitates immune evasion by *Pg* and facilitates its intracellular survival, ultimately promoting the development of chronic inflammation in arteries [75].

*Pg* can suppress the proliferation of vascular ECs and induce cell apoptosis, thus destroying the endothelial barrier [76]. It is evidenced that gingipains can cleave neural cadherin and vascular endothelial cadherin, and degrade integrin β1, making ECs disconnect from the ECM, and come to anoikis [77,78]. Systemic inflammation induced by *Pg* can also promote the endothelial–mesenchymal transition (EndMT) of ECs, thereby promoting the fibrosis of arterial plaques and destroying the permeability and integrity of the vessel wall [65]. *Pg*-activated ECs secrete angiotensin II and pro-inflammatory cytokines such as interleukin (IL)-6, monocyte chemoattractant protein-1 (MCP-1), and granulocyte-macrophage colony-stimulating factor (GM-CSF), amplifying vascular inflammation and arterial hypertension [79].

Vascular oxidative stress is one of the pathological mechanisms of many cardiovascular diseases, including AS, hypercholesterolemia, hypertension, and diabetes mellitus [80]. Excessive or sustained reactive oxygen species (ROS) are the main characteristic of oxidative stress. *Pg* can promote ROS production in multiple ways. It is identified that *Pg* can motivate DNA methyltransferase 1 (DNMT-1) to methylate basic helix-loop-helix ARNT like 1 (BMAL1) promoter via activating TLRs-NF-κB signal axis, followed by the restrain of BMAL1 expression and release of circadian locomotor output cycles kaput (CLOCK). In turn, CLOCK phosphorylates P65 and further enhances the NF-κB signal, which aggravates oxidative stress and inflammatory response in human aortic ECs, thereby aggravating vascular endothelial injury and promoting the progress of AS [46]. *Pg* is also able to reduce the antioxidant mechanism and accelerate the oxidative damage of ECs through the NOS/BH4/Nrf2/GSK-3β pathway [81]. It is known that excessive ROS accumulation, mitochondrial oxidative stress damage, disrupted mitochondrial dynamics, and an inadequate energy supply caused by mitochondrial dysfunction can aggravate AS. A recent study reveals that *Pg* induced mitochondrial fission and dysfunction in ECs by promoting the phosphorylation and recruitment of dynamin-related protein 1 (Drp1), an indicator of mitochondrial fission, leading to the increase in mitochondrial ROS, as well as the decrease in mitochondrial membrane potential and ATP [82], which threaten the integrity of vascular endothelial.

#### 2.2.2. Monocyte Adherence and Aggregation

Monocyte recruitment to the endothelium is a crucial step in AS. Endothelium injury causes the subsequent chemotaxis and aggregation of monocytes to the subendothelium. *Pg* upregulates the expression of MCP-1, ICAM-1, VCAM-1, and E- selectin in ECs, and the expression of C-C chemokine receptor 2 (CCR2) and integrinαMβ2 in monocytes, promotes the adhesion and aggregation of monocytes to the endothelium [83,84,85,86,87]. The NF-κB pathway plays a vital role in this process. Restraint of the NF-κB pathway can abrogate ICAM-1 expression in ECs [88]. NOD1, an intracellular pattern recognition reporter, is overexpressed in *Pg*-infected ECs. After NOD1 recognizes *Pg*, the expression of ICAM-1 and VCAM-1 in ECs is up-regulated through the NF-κB signaling pathway [84]. Gas6 inhibits *Pg*-LPS-induced monocyte–endothelial cell interaction in vitro through the Akt/NF-κB pathway [89]. Macrophage migration inhibitory factor (MIF) secreted is augmented in *Pg*-infected ECs, which binds to CD74 and CXCR4 on the surface of ECs to form a receptor–ligand complex and activates ECs to express more ICAM-1 [90]. Exposure to *Pg* induces the increased expression of these adhesion molecules and attracts a large number of monocytes to accumulate in the subendothelium of the artery, which results in extensive secretion of inflammatory factors and exacerbates vascular and systemic inflammation [65,91]. Cyclic diadenylate monophosphate (c-di-AMP) can mitigate the effect of *Pg* on AS by activating trained immunity and regulating microecological balance, including relieving the elevation in gene expression of IL-6, IL-1β, TNF-α, and interferon β; ECM remodeling enzymes matrix metalloproteinase (MMP)-2 and MMP-9; and adhesion molecules ICAM-1 and VCAM-1 [92].

#### 2.2.3. Foam Cell Formation

The formation of foam cells is a hallmark of AS. Macrophages serve as one of the primary sources of foam cells in plaque. *Pg* and its components, including outer membrane vesicles (OMVs), can boost the binding and phagocytosis of macrophages to low-density lipoprotein (LDL), and macrophage-mediated modification of LDL [93]. *Pg* can increase the expression of CD36, a scavenger receptor that mediates cholesterol uptake through the c-Jun-AP-1 pathway [94] or ERK/NF-κB [95], as well as lysosomal integral membrane protein 2 (LIMP2) involved in cholesterol transport [96,97], so as to intensify the lipid accumulation of macrophages. *Pg*-infected macrophages up-regulate fatty acid binding protein 4 (FABP4), an intracellular transport protein for fatty acids, presenting more intake of fatty acids [98]. Moreover, a notable positive association was observed between serum *Pg* antibody and FABP4 level in clinical periodontitis patients, suggesting that *Pg* can promote AS and other systemic diseases by affecting FABP4 [98]. *Pg* also enhances the activity of calpain of macrophages in a dose-dependent manner, increases the degradation of ATP-binding cassette transporter A1 (ABCA1), and subsequently hinders the excretion of cholesterol [94], contributing to foam cell formation of macrophages. Additionally, *Pg* also boosts the oxidation of high-density lipoprotein (HDL), impairing its reverse cholesterol transport function and shifting its role from a protective factor against AS to a pro-atherogenic factor [99].

#### 2.2.4. Calcification and Angiogenesis in Plaque

In the advanced stages of AS, the presence of calcium deposition within plaques, known as calcification, is frequently observed. Calcified plaques contribute to luminal narrowing and impede blood flow. However, it is worth noting that there is a prevailing viewpoint suggesting that calcified plaques exhibit greater stability and reduced susceptibility to rupture compared to non-calcified plaques. VSMCs also constitute a large portion of the plaque. *Pg* promotes phenotypic transformation, apoptosis, and matrix vesicle release of VSMCs, and consequently intensifies inorganic phosphate-induced vascular calcification [100]. *Pg* boosted VSMCs proliferation and intimal hyperplasia, and the expression of vascular cell proliferative phenotypic markers S100 calcium-binding protein A9 (S100A9) and embryonic isoform of smooth muscle myosin heavy chain (SMemb) was observed higher on the surface of VSMCs of *Pg*-infected mice and in aneurysm specimens from *Pg*-infected patients [101,102]. Microarray analysis suggested that *Pg* may promote VSMCs proliferation through 25 pathways, including the transforming growth factor-β (TGF-β), Notch, MAPK, ErbB, calcium signaling pathways, and so on [103]. OMV of *Pg* up-regulates runt-related transcription factor 2 (Runx2) via ERK1/2 signaling, which drives SMC differentiation and calcification activation, and eventually exacerbates vascular calcification [104]. In one study where human VSMCs and periodontal ligament cells were co-cultured, VSMCs presented a higher degree of calcification under the stimulation of *Pg*-LPS than those cultured alone, which indicated the effect of periodontitis on vascular calcification [105].

In response to the oxygen and nutrient demands, new blood vessels gradually form within the growing plaque, while they are often structurally abnormal and fragile. Angiogenesis may cause leakage of blood cells and inflammatory cells into the plaque, or even rupture and lead to intraplaque hemorrhage. Microarray analysis revealed that gingipains influence the focal adhesion activation, ECM receptor interactions, and the actin cytoskeleton pathway of *Pg*-mediated VSMCs, suggesting an impact on VSMC motility, phenotype transition, and angiogenesis processes [106]. *Pg* and its gingipains have been demonstrated to facilitate the upregulation of the high angiopoietin 2 (Angpt2)/Angpt1 expression ratio in VSMCs, manifesting their potential involvement in promoting vascular neogenesis, SMC proliferation, and pro-inflammatory phenotypic changes. Conversely, fimbriae and LPS lack the ability to elicit similar effects [107]. Furthermore, *Pg* can stimulate VSMCs to generate tissue factor (TF) pathway inhibitor, which serves as the primary endogenous inhibitor responsible for regulating the TF-mediated blood coagulation cascade, and may promote the occurrence of acute coronary symptoms [108].

#### 2.2.5. Plaque Destabilization

There is currently a dearth of research regarding the association between *Pg* and plaque destabilization in AS. A study conducted in 2017 demonstrated that *Pg* can facilitate the imbalance between Th17 and Treg cells, and encourage intra-plaque inflammation by modulating T cell differentiation during the progression of AS. This contributed to an enlarged AS lesion area, accompanied by an escalation in macrophage content and a reduction in VSMC area, thereby fostering plaque instability [63]. *Pg* also triggered macrophages to secret MMP9, thereby inducing the fragmentation of vascular type IV collagen, which weakened the structural support of the plaque and worsen its destabilized [109]. Be, the heightened vascular inflammation, also impairs plaque stability [110]. Moreover, Hgp44, the bacterial surface adhesin of *Pg*, which is cleaved by Arg-specific gingipain (Rgp) and Kgp, can interact with circulating reactive IgG against *Pg*, Fc γRIIa receptors, and GPIb α receptors on the platelet surface, leading to platelet aggregation and escalating the risk of AS-related cardiovascular events [111] (Figure 1) (Table 1).

## 3. *Aggregatibacter actinomycetemcomitans*

*Aa*, a Gram-negative facultent-anaerobic coccobacillus, is the predominant bacterium isolated from caries in adolescents and adults with stage Ⅲ or Ⅳ periodontitis [112] and can lead to premature tooth loss. *Aa* is also one of the major bacteria in the subgingival biofilm of patients with periodontitis [50]. Ramin Akhi et al. observed that the levels of salivary IgA antibodies to MAA-LDL (*p* = 0.034) and *Aa*-HSP60 (*p* = 0.045) increased with an elevated number of teeth with probing depths of 4–5 mm, which may suggest the cross-activation of the humoral immune may potentially mediate the association between PD and systemic disorders [113]. Pili is an important structure for *Aa* adhesion to the host. Isolated *Aa* pili contained a low molecular mass protein (about 6.5 kDa), called Flp, and a small amount of a 54-kda protein, called Fup [114]. Examination of the binding of Aa to hydroxyapatite surfaces coated with saliva exhibited a highly adhesive interaction that seemed to rely on the formation of glycoconjugates [115]. In an oral colonization model infected with the Flp mutant of *Aa*, the absence of soft tissue or plaque colonization, as well as the absence of bone loss, in the Flp mutant of Aa, provides compelling evidence supporting the critical role of Flp in *Aa*’s virulence [116].

### 3.1. The Association between Aa Exposure and AS in Humans and Animals

Epidemiological investigations have shown a significant association between *Aa* and coronary disease. According to a meta-analysis conducted in 2022, the prevalence of *Aa* in clinical coronary atherosclerotic plaques was found to be 46.2% (95% CI: 20.6–74.0) [117]. Kozarov E. et al. revealed that the prevalence of *Aa* positivity in plaques among elderly AS patients is approximately 55.5% [118]. Serologic evidence described that rheumatic arthritis (RA) patients with *Aa* or *Aa* leukotoxin (LtxA) exposure has higher coronary artery calcification, thicker IMT, and lower ankle-brachial index (ABI, a surrogate for peripheral AS), indicating greater progression of AS [119]. In the context of ruptured cerebral aneurysms, the detection rate of *Aa* is notably higher, estimated at around 14% [120]. A Finnish study in 2012 demonstrated that after adjusting for CVD risk factors, a 10-fold increase in *Aa* level of saliva was associated with an odds ratio (OR) of 7.47 (95% CI = 1.57–35.5, *p* = 0.012) for stable CAD, and an OR of 4.31 (95% CI = 1.06–17.5, *p* = 0.041) for ACS [121]. As well, serum IgA levels targeting *Aa* exhibited a correlation with ACS (OR = 3.13, 95% CI = 1.38–7.12, *p* = 0.006) [121]. Molecular mimicry results indicate a potential effect of *Aa* in the pathogenesis of AS through the activation of cross-reactive immune responses [122]. *Aa* possesses multiple serotypes. On the basis of structural features of the LPS o antigen, seven serotypes of *Aa* (“a”–“g”) have been identified [123,124,125,126]. Statistically, serotypes “b” and “c”, which are linked to periodontal probing depth and the burden of periodontal inflammation, happen to be the most common serotypes in patients with CAD (59.3%) (*p* = 0.040) and with the severity of CAD [127]. In *ApoE*^-/-^ mice, both *Aa* and *Aa* LPS can promote AS progression and plaque lipid deposition, and the virulence ranked from highest to lowest was *Aa* > heat-inactivated *Aa* > *Aa* LPS [128]. *Aa* infection can alter the serum lipoprotein profiles with augmented very low-density lipoprotein (VLDL), LDL, and HDL [129]. Oxidized LDL (ox-LDL) levels and the expression of oxidative stress-related molecules such as myeloperoxidase (MPO), the receptor for advanced glycation endproducts (RAGE), inducible nitric oxide synthase (iNOS), NADPH oxidase-related genes *Nox-1*, *Nox-2*, and *Nox-4* are also increased after *Aa* administration, indicating that *Aa* plays a central role in oxidative stress and oxidation of low-density lipoprotein [128]. It has also been proposed that *Aa* did not affect the serum cholesterol level and the expression of hepatic fatty acid synthase (FAS) and HMG CoA reductase in *ApoE*^-/-^ mice, but the inflammatory factors and chemokines in serum, innate immune signaling molecules such as TLR2 and TLR4, adhesion molecules (ICAM-1, E-selectin, and P-selectin), chemokines (C-C motif chemokine ligand (CCL)19, CCL21, CCR7, and MCP-1), scavenger receptors lectin-like oxidized low-density lipoprotein receptor-1 (LOX-1) and high-sensitive C-reactive protein (hsCRP) in the aorta were significantly upregulated compared with the control group [130].

### 3.2. Mechanistic Investigations on the Effect of Aa in AS

*Aa* is known to generate two types of toxins, LtxA and cytolethal-distending toxins (Cdts). LtxA binds with lymphocyte function-associated antigen-1 (LFA-1; CD11a/CD18), the receptor for LtxA on leukocytes [131,132,133], to induce macrophages pyroptosis and activation of inflammasome to release inflammatory cytokines and induce secondary immune response [134]. *Aa* has a significant impact on inflammation as well as the aggregation and adhesion of monocytes by up-regulating ICAM-1 and VCAM-1 on ECs [135] and macrophages [136,137]. LtxA can also arrest the G2/M phase of the cell cycle in microvascular ECs, thus hindering cell proliferation and driving cell apoptosis [135]. LPS of *Aa* can induce TNF-α and IL-1β production, and restrain the expression of scavenger receptor class B type-I (SR-BI) and ABCA1 in macrophages, followed by augmented cholesterol accumulation [138]. Infection with *Aa* elevates serum and intramural levels of TH17 cell-related factors, such as IL-1β, IL-17, IL-6, TGF-β, and IL-1β, indicating a potential induction of TH17 activation and the promotion of vascular inflammation [139]. This evidence illustrates a certain correlation between *Aa* and CVD. Cdts, a heterotrimeric AB2 toxin, can be internalized into cells and induce cell-cycle retardation and apoptosis in lymphocytes and other cell types [140]. However, the research on the role of Cdts and other virulence factors of *Aa* in AS is still inadequate (Figure 2) (Table 2).

## 4. *Fusobacterium nucleatum*

*Fn* is a species of bacteria that belongs to the genus Fusobacterium. It is a Gram-negative anaerobic bacterium. *Fn* acts as a bridge bacterium, facilitating the adherence of other bacteria to form complex microbial communities. It exhibits various virulence factors that contribute to its pathogenicity, including adhesins such as adhesin FadA and Fap2 [141], outer membrane proteins like radial proteins D [141], hemagglutinins, secreted toxins like butyric acid [142], LPS, and some proteases.

### 4.1. The Association between Fn Exposure and AS in Humans and Animals

In recent years, considerable attention has been directed toward the involvement of *Fn* in extra-oral infections, encompassing conditions such as colorectal cancer [143] and RA [144]. A substantial body of research has established a link between *Fn* and CVD [145,146]. In a study involving 31 resected carotid endarterectomy specimens, the positivity rate of *Fn* detection was found to be 34% [147]. It has been frequently identified within atherosclerotic plaques, and it stands as one of the predominant periodontal pathogens detected in ruptured cerebral aneurysms [120]. The presence of *Fn* in atherosclerotic plaques and blood vessels increases proportionally with the severity of periodontal disease [148]. It is shown that subjects with stable CAD exhibit elevated levels of subgingival-specific IgA targeting *Fn* [149], compared to subjects without significant coronary stenosis. In *ApoE*^-/-^ mice, oral administration of *Fn* significantly increased plasma triglyceride (TG) and cholesterol levels, facilitating the development of AS, and inducing a transition of plaques toward an unstable phenotype [150,151]. GroEL is a heat shock protein produced by *Fn*. *ApoE*^-/-^ mice injected with GroEL exhibited elevated serum levels of AS risk factors such as IL-6, CRP, and LDL, while the concentration of HDL decreased [152].

### 4.2. Mechanistic Investigations on the Effect of Fn in AS

Recent studies have elucidated that *Fn* can enhance EC permeability and reduce the abundance of EC adhesion molecule-1, leading to endothelial dysfunction [153]. *Fn* has been shown to impair ECs proliferation and induce apoptosis [154,155]. *Fn* and its GroEL *Fn* are capable to upregulate the expression of chemotactic factors, including MCP1 and IL-8, as well as cell adhesion molecules including ICAM-1, VCAM-1, and E-selectin in ECs [152]. *Fn* also disrupts lipid metabolism and transport processes. *Fn* fosters hepatic glycolysis and lipid synthesis through the PI3K/Akt/mTOR signaling pathway, thus uplifting plasma lipid concentrations and exacerbating AS in mice [150]. *Fn*-infected macrophages exhibit an activation of the PI3K-AKT/MAPK/NF-κB signaling pathway, propelling the inflammatory responses and cholesterol uptake, concurrently reducing lipid excretion, leading to lipid deposition [156]. Another study revealed that *Fn* stimulates macrophages to undergo abnormal pro-inflammatory responses, M1 polarization, and cell apoptosis, triggering the release of inflammatory cytokines such as IL-6, IL-1β, IL-17, and TNF-α. This process, in turn, modulates the expression of lipid metabolism-related genes, including *scavenger receptor A1* (*SR-A1*), *cholesterol acyltransferase 1* (*ACAT-1*), *ABCA1*, and *ATP-binding cassette transporter G1* (*ABCG1*), promoting the accumulation of cholesterol in macrophages induced by ox-LDL [151]. In addition, *Fn* upregulates FABP4 via a JNK-dependent mechanism in macrophages, thereby augmenting lipid uptake by macrophages [98] (Figure 3) (Table 3).

## 5. *Prevotella intermedia*

*Pi*, a Gram-negative bacterium, is a dominant bacterium of periodontitis and is predominant in adult patients with periodontitis [157,158]. Two genotypes, I and II, have been identified for *Pi* [159]. In 1992, Harou N N Shan et al. identified a new genetic group in the *Pi* strain, which was significantly different from genotype I in terms of DNA–DNA hybridization characteristics and peptidase and lipase activities. The new species was named *Prevotella nigrescens* (*Pn*) [160]. *Pi* and *Pn* can simultaneously exist in oral mucosa, the tongue, and tonsils, as well as in subgingival plaque in deep periodontal pockets [161,162]. Some studies have proposed that *Pn* strains are associated with healthy sites, whereas *Pi* strains are isolated from deeper sites of periodontal lesions and are thought to connect with periodontal breakdown [163,164,165]. In addition to periodontitis, *Pi* is also found to be abundant in colorectal cancer [166]. Moreover, *Pi* has been linked to subclinical hypothyroidism [167], infectious endocarditis [168], and other related conditions.

### 5.1. The Association between Pi Exposure and AS in Humans and Animals

It was observed the positive rate of *Pi* in the atherosclerotic artery was found to be 31% [47]. A multiple regression model adjusting for known cardiovascular risk factors illustrated that the level of IgA to Pi in patient saliva and IgA against LDL were significantly associated with stable CAD and ACS [169]. Multivariate logistic regression analysis displayed that *Pn* infection was linked to increased cross-sectional intimamedia area (OR = 4.08; *p* = 0.034) [170]. In smokers, *Pi* antibody levels were significantly related to CHD (OR = 1.5), while in never-smokers, it was *Pn* that was associated with CHD (OR = 1.5) [171]. Serum anti-*Pi* antibodies showed a significant correlation with the internal carotid artery (ICA) AS (OR = 16.58, 95% CI = 3.96–78.93, *p* < 0.0001) [172]. After excluding ICA AS, a strong relationship between anti-*Pi* antibodies and atherosclerotic stroke was observed (OR = 23.6, 95% CI = 2.65–298.2, *p* = 0.008) [172]. Despite the lack of straight evidence for the relevance between *Pi* and AS, some studies suggest a role for *Pi* in AS-associated cells.

### 5.2. Mechanistic Investigations on the Effect of Pi in AS

Currently, research on the impact of *Pi* on macrophages is limited. The pro-inflammatory effect of Pi on macrophages may aggravate the progression of AS. LPS is the major virulence factor of Pi. Similar to the LPS of *Pg*, the LPS of Pi differs greatly in the structure from LPS of Escherichia coli. Pi-LPS contains fewer and longer fatty acids than *E. coli*-type lipid A [173]. Pi-derived LPS can attract the production of macrophage inflammatory mediators such as nitric oxide (NO), IL-1β, and IL-6 through TLR4 signaling pathway [174]. In addition, a novel non-endotoxin protein, prevotella glycoprotein, was isolated from *Pi*, which is composed of carbohydrates and protein and is free of fatty acids [175]. PCG raises IL-8 production by human monocyte THP-1 cells and motivates human and mouse monocytes through CD14 and TLR2 but not TLR4-dependent pathways [176]. For other cells associated with AS, there are in vitro invasion assays indicating that *Pi* can invade and infect primary cultures of human coronary artery ECs and coronary artery VSMCs [177]. In the future, more studies are needed to reveal the effect of *Pi* on AS and its mechanism (Table 4).

## 6. *Tannerella forsythia*

*Tf* is an anaerobic Gram-negative member of the *Cytophaga-Bacteroides* family. It was first isolated by the Forsyth Institute in the 1970s from subjects with advanced progressive periodontitis, and originally described as Fusiform Bacteroides [158]. According to the 16S rRNA phylogenetic analysis, it was reclassified to *Tf* [189,190]. *Tf* is one of the members of the “red complex”, and participates in the development of gingivitis and periodontitis. Multiple research studies have successfully detected *Tf* in the subgingival biofilm of patients using various techniques [50]. Several potential virulence factors have been discovered in *T. forsythia*, including trypsin-like [191] and PrtH proteases [192], NanH [193], a leucine-rich repeat protein BspA [194], alpha-D-glucosidase, N-acetyl-beta-glucosaminidase [195], components of the bacterial S-layer, and methylglyoxal [196].

### 6.1. The Association between Tf Exposure and AS in Humans and Animals

In a study, the detection rate of *Tf* in the atherosclerotic artery was remarkably high, reaching 53% [47]. Patients with CAD and ACS demonstrated a pronounced elevation in saliva levels of IgA to *Tf* relative to individuals without CAD [169]. In animal models of AS, *Tf* and BspA promote plaque enlargement and increase serum levels of CRP and LDL, while the level of HDL, and the expression of liver X receptor (LXR)α, LXRβ, and ABCA1 in liver tissue of AS mice are lessened [181]. Oral infection of *Tf* significantly lowers the serum NO level in mice, and increases the degree of serum acute phase reactant SAA, which is a subclinical inflammatory marker of chronic disease characterized by inflammation [182]. *Tf* is also significantly connected with intraplaque hemorrhage and plays a potential role in neutrophil activation within hemorrhagic carotid plaques, which ultimately exacerbates plaque vulnerability [183].

### 6.2. Mechanistic Investigations on the Effect of Tf in AS

The mechanisms underlying the involvement of *Tf* in the promotion of AS remain inadequately understood. Existing research findings indicate that *Tf* and its components, including LPS and OMVs, can enhance the secretion of pro-inflammatory mediators by macrophages, including IL-1β, IL-6, TNF-α, and IL-8 [178,179,180], while S-layer-deficient *Tf* mutants yield a remarkably higher secretion level [179]. Similarly, infection of mice with *Tf* mutant strains lacking an intact S-layer glycan core has been shown to provoke robust Th17 cell responses and researchers considered that the surface glycosylation of *Tf* may contribute to its persistence within the host by restraining Th17 responses [197]. *Tf* and BspA can also induce THP-1 to form foam cells [181]. This evidence manifests that *Tf* may evade recognition by the innate immune system, elicit a chronic inflammatory response, and catalyze foam cell formation in AS plaque (Table 4).

## 7. *Treponema denticola*

*Td* is a Gram-negative bacterium from the Spirochetes family. As a partner of the “red-complex” organisms, *Td* is commonly found in the oral cavity, especially in subgingival plaque. *Td* possesses several virulence factors, such as the major outer sheath protein (MSP), ortholog of oligopeptide transporter unit (OppA), factor H-like protein-1 binding proteins, coaggregation, dentilisin, lipooligosaccharide [198], peptidoglycan, and cystalysin, which assist *Td* in adhesion, locomotion, immune escape and destruction of host cells [199].

### 7.1. The Association between Td Exposure and AS in Humans and Animals

PCR testing of vascular plaques showed that the detection rate of *Td* is 6%, which is the lowest among periodontal bacteria [47]. In former smokers, *Td* antibody titers are higher and significantly associated with CHD (OR = 1.7; 95% CI = 1.2–2.3) [171]. In the context of stage Ⅲ or Ⅳ periodontitis, the presence of *Td* was found to be linked with elevated levels of TG and reduced levels of HDL (OR = 3.03; 95% CI = 1.2–7.2) [186]. Sasanka S. Chukkapalli et al. discovered that mice orally infected with *Td* had enlarged arterial plaques, decreased serum NO levels, and increased serum VLDL and ox-LDL levels [185].

### 7.2. Mechanistic Investigations on the Effect of Td in AS

*Td* can activate human ECs by inducing IL-8 and MCP-1 expression [184], which facilitates the chemotaxis and aggregation of monocytes to the subendothelium (Table 4).

## 8. *Campylobacter rectus*

*Cr* is a Gram-negative anaerobic bacterium. It was initially named and identified in 1981 as *Wolinella recta*, and was reclassified as *Campylobacter* in 1991 based on phylogenetic analysis [200]. It is common in the oral cavity and gastrointestinal tract and mainly participates in oral and periodontal infections, but it is also detected in cases of severe infection outside the gastrointestinal tract [201].

### 8.1. The Association between Cr Exposure and AS in Humans and Animals

Subjects with CAD showed elevated levels of subgingival *Cr*-specific IgA compared to individuals without significant coronary stenosis (IQR = 1.20; *p* = 0.009), and the difference was pronounced in the group with ACS (IQR = 1.16; *p* = 0.050) [149]. Elevated antibody titer to *Cr* was linked to thick carotid arterial walls (OR = 2.3, 95%CI = 1.83–2.84) [188].

### 8.2. Mechanistic Investigations on the Effect of Cr in AS

Similar to other oral bacteria, *Cr* possesses potent TLR4 stimulating activity, effectively triggering the macrophage TLR4 signaling pathway and inducing IL-6 secretion [187]. Currently, there is a lack of evidence of how *Cr* affects AS, and further research is needed (Figure 4) (Table 4).

## 9. Interactions between Bacteria

In practical scenarios, infections commonly involve polymicrobial communities rather than single bacterial species. Interactions among different periodontal pathogens play a pivotal role in the pathogenesis of these infections. For instance, *Pg* primarily utilizes gingipains to aggregate with *Tf*, facilitating macrophage-mediated engulfment and clearance of *Tf*, while mutation of either Kgp or Rgp in the coinfecting *Pg* resulted in a diminished enhancement of *Tf* phagocytosis [202]. The invasive capacity of *Pg* significantly increases when co-cultured with *Fn* compared to *Pg* incubated alone, possibly because *Fn* facilitates *Pg* adhesion to cells [203]. *Cr* can provide a protoheme for *Pg* growth [204]. Most periodontal pathogens primarily belong to Gram-negative bacteria and exhibit common features such as LPS and OMV, which can activate inflammatory responses. Meanwhile, they possess unique virulence factors contributing to their pathogenicity. Indeed, similar to periodontitis, the complex interplay of competition and cooperation among different microorganisms shapes the property and functionality of the oral biofilm. Interspecies signaling within the community regulates various activities, including gene expression, nutrient acquisition, and DNA exchange, enabling the bacterial consortium to exhibit multifaceted microbial synergistic interactions [205]. It has been established in animal studies that polymicrobial infections may intensify tissue destruction in comparison to single-species infections. For instance, co-infection of *Pg* and *Tf* results in more pronounced bone loss and inflammatory response on the surfaces of the maxilla, mandible, palate, and cheeks in rats, surpassing the effects observed with individual *Pg* or *Tf* infections [206]. The co-infection of *Fn* and *Tf* has been shown to significantly increase the infiltration of inflammatory cells in gingiva and heighten osteoclast activity in the alveolar bone [207]. Mice infected with multiple periodontal bacteria exhibits marked alterations in serum levels of AS-related molecules [208,209]. HIV-infected individuals co-infected with hepatitis C virus face an elevated risk of CVD and the occurrence of atherosclerotic CVD events [210,211]. In future investigations, it is crucial not only to elucidate the specific mechanisms by which each pathogenic bacterium contributes to AS, but also to explore the collective impact of polymicrobial infections on AS pathology.

The oral mucosa and intestinal mucosa exhibit a physical continuity, establishing a direct link between the oral microbiota and gut microbiota [212]. Approximately 1.5 × 10^12^ oral bacteria are ingested daily through saliva swallowing [213]. Oral microbiota can colonize the intestine, forming a dual microbial ecosystem and the immune mechanism that connects oral and gut health [214], which provides a potential pathway for PD to extend its impact systemically through the gastrointestinal tract. Accumulating evidence indicates the active participation of the oral–gut axis in the pathogenesis of several diseases; for example, a cross-sectional study observed compositional changes in the oral and gut microbiota of hypertensive patients, as well as that certain common periodontal bacteria undergo ectopic colonization in the intestines and saliva-derived *Veillonella* exhibits the most significant variations in abundance [215]. Animal investigations have further corroborated a causal relationship between oral–gut microbial transmission and the exacerbation of hypertension [215]. It has been implicated that oral pathogens and their byproducts, along with inflammatory mediators, can cause dysbiosis in the intestinal ecosystem, gaining access to the systemic circulation through compromised intestinal barrier, and aggravating liver cirrhosis [216]. The oral microbiota associated with periodontitis exacerbates intestinal inflammation and compromises the integrity of the intestinal barrier in PAP mice. This disruption leads to dysregulation of the gut microbiota, which worsens the progression of Alzheimer’s disease through the intricate gut–brain crosstalk in mouse models [217].

A substantial body of research has been conducted to explore the respective roles of oral microbiota and gut microbiota in AS; however, the involvement of the oral–gut axis in AS is less understood. Comparative analysis of microbial populations in AS plaques, oral cavity, and intestines has revealed notable similarities in microbial abundance between AS plaques and the oral cavity, while the intestines exhibit the highest microbial richness [218]. Remarkably, several bacterial taxa, such as *Firmicutes*, *Clostridia*, *Veillonella*, *Streptococcus*, and so on, are found to be shared in these anatomical sites [218]. The established influence of the gut microbiota on AS involves several mechanisms, such as the activation of inflammatory pathways, modulation of lipid metabolism, and generation of specific metabolites [219,220,221]. The compromised gut barrier and dysbiosis caused by periodontitis may participate in the progression of AS through these mechanisms. In a recent study conducted by Guowu Gan et al., it was discovered that chronic apical periodontitis induced by *Pg* infection in mice resulted in significant alterations in the composition and diversity of the gut microbiota. Specifically, they observed significant differences in the abundance of 37 taxa, wherein an increase in the relative abundance of *Firmicutes*, *Chloroflexi*, and *Cyanobacteria*, along with a decrease in *Bacteroidetes*, correlated with the progression of AS (*p* < 0.05) [222]. In the context of chronic apical periodontitis, *ApoE*^-/-^ mice exhibited elevated serum levels of trimethylamine N-oxide (TMAO), a gut-derived metabolite that indicates the risk of cardiovascular events [223]. Co-occurrence analysis demonstrated a significant positive correlation (*p* < 0.05) between the relative abundances of *Lachnospiraceae* and *Porphyromonadaceae* with both the percentage of lesion area and TMAO levels, suggesting that chronic apical periodontitis exacerbates atherosclerotic lesions by modulating the gut microbiota and its metabolites [223]. The aforementioned findings imply that the gut microbiota may serve as a pivotal link between oral infection and the pathogenesis of AS. The oral–gut axis holds intrinsic potential in the development of AS and in-depth explorations are needed. Microbiota status holds promising prospects for the identification of potential indicators for disease risk assessment and prognostic evaluation, as well as novel therapeutic strategies. Compared to blood biochemistry testing, the detection of samples from the oral cavity and gut offers a more non-invasive and convenient approach. Moreover, the development of strategies aimed at manipulating the diversity of oral and gut microbiota, such as targeted oral hygiene protocols, dietary nutrition interventions, and the administration of prebiotics, may hold promise for the prevention and treatment of AS.

## 10. Conclusions

In conclusion, this comprehensive review primarily summarizes the epidemiological evidence linking periodontal pathogens to AS and highlights the existing research on the mechanisms by which periodontal bacteria contribute to the progression of AS. It emphasizes that periodontal pathogens are important risk factors for AS and underscores the need for further research in this area, as there is still a lack of comprehensive understanding. Furthermore, it raises awareness of the complex interactions exhibited by periodontal bacterial co-infections and the oral–gut axis in the context of AS.

At last, although the evidence linking periodontal pathogens to AS is accumulating, the current body of evidence is inadequate to establish the effectiveness of periodontal treatment in preventing or ameliorating AS. Future studies should also concentrate on investigating the potential impact of periodontal interventions on the prevention and management of AS. Moreover, it should be noted that the current research on the relationship between periodontal bacteria and AS primarily relies on conventional detection methods. However, the application of innovative techniques such as single-cell sequencing and spatial omics in this field remains largely unexplored. The integration of these advanced methodologies is expected to facilitate a more comprehensive understanding of the role of periodontal bacteria in the cellular and intercellular interactions within the AS microenvironment [97]. Consequently, this will provide crucial theoretical foundations for the prevention and treatment of both periodontal disease and atherosclerosis.

## Figures and Tables

**Figure 1 ijms-24-12861-f001:**
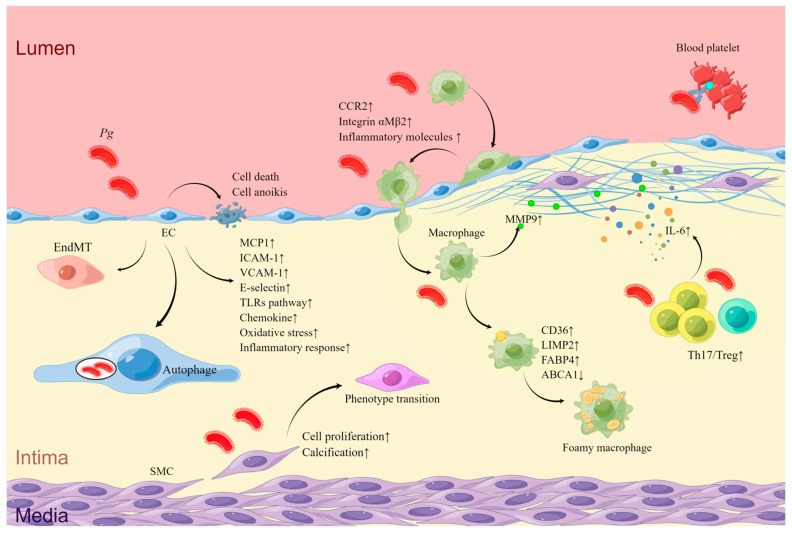
The mechanisms of *Pg* in AS. *Pg* infection leads to upregulation of adhesion molecules and chemokine secretion in ECs [70,71,72,84,88,89,90], promoting adhesion and aggregation of monocytes in the bloodstream [83,84,85,86,87]. *Pg* can activate autophagy in ECs, allowing it to evade immune responses by residing in autophagic vesicles [73,74]. *Pg* also inhibits EC proliferation [76], promotes EC apoptosis [75,76,77,78], epithelial–mesenchymal transition [65], oxidative stress, and inflammatory factor secretion [46,81,82], ultimately disrupting the vascular endothelial barrier. Under *Pg* stimulation, macrophages produce more inflammatory mediators [92], leading to increased lipid deposition [93,94,95,96,97,98,99] and exacerbating the inflammatory environment within the plaque. *Pg* promotes proliferation [103,107], phenotypic transformation [100,106,107], and calcification [104] of VSMCs. *Pg*-induced imbalance in the differentiation of Th17 and Treg cells worsens inflammation within the plaque [63]. Increased secretion of MMP9 by macrophages promotes collagen degradation and plaque instability [109]. *Pg* promotes platelet aggregation and thrombus formation by interacting with IgG and Fc γRIIa receptors [111] (drawn by Figdraw).

**Figure 2 ijms-24-12861-f002:**
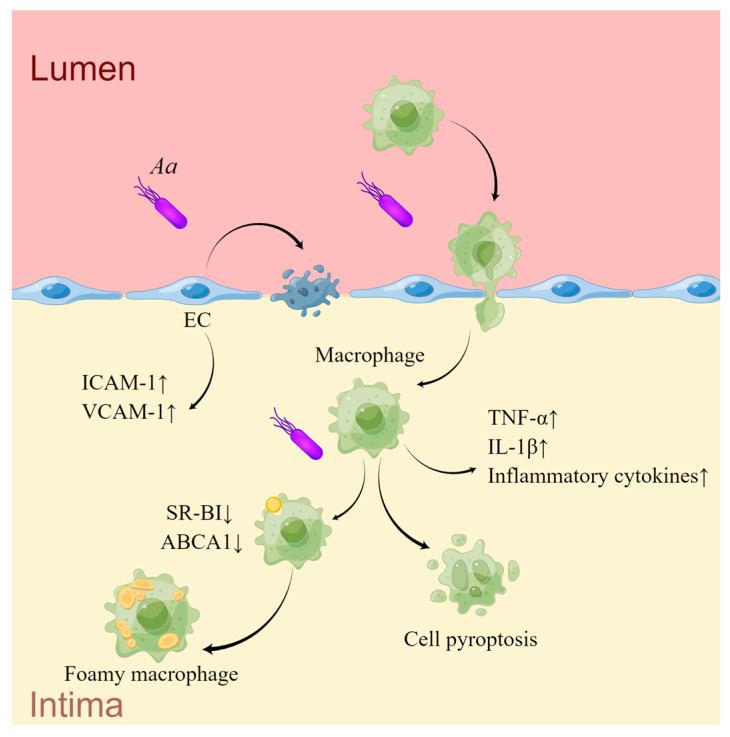
The mechanisms of *Aa* in AS. *Aa* infection upregulates the expression of adhesion molecules ICAM-1 and VCAM-1 in ECs, promoting adhesion and aggregation of monocytes to subendothelium. *Aa* also induces apoptosis in EC, leading to disruption of the vascular endothelial barrier. Under *Aa* stimulation, macrophages exhibit increased secretion of inflammatory cytokines and foam cell formation. *Aa* also promotes macrophage pyroptosis (drawn by Figdraw).

**Figure 3 ijms-24-12861-f003:**
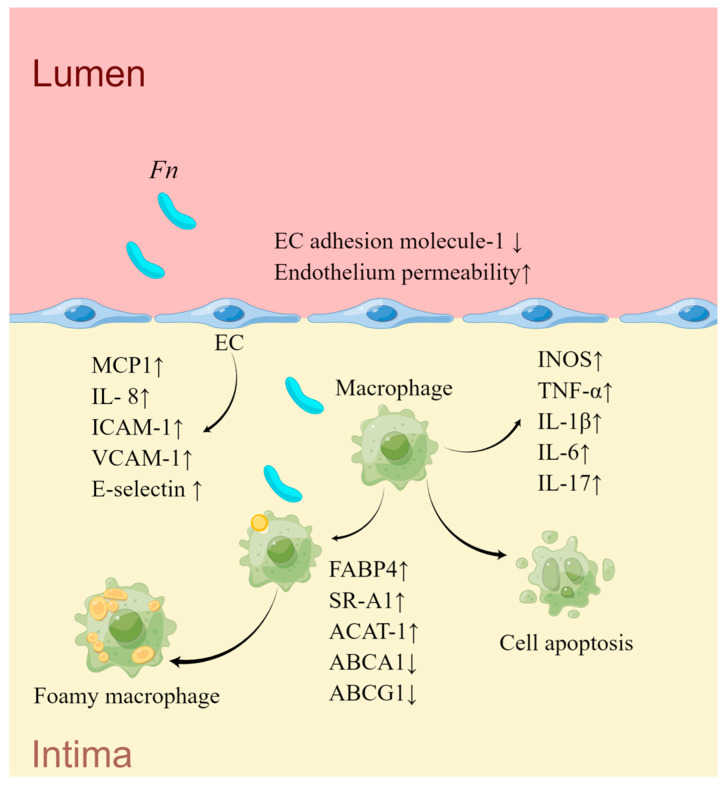
The mechanisms of *Fn* in AS. *Fn* infection disrupts the intercellular connections between EC, simultaneously inhibiting EC proliferation and promoting EC apoptosis, leading to increased endothelial permeability. *Fn* also upregulates the expression of adhesion molecules and chemokines in EC, promoting the adhesion and migration of monocytes toward the sub-endothelial space. Under *Fn* stimulation, macrophages undergo M1 polarization, resulting in increased production of inflammatory mediators. Macrophage lipid deposition and cell apoptosis are also heightened, fostering the progression of plaque enlargement and an inflammatory environment within the plaque. (Draw by Figdraw).

**Figure 4 ijms-24-12861-f004:**
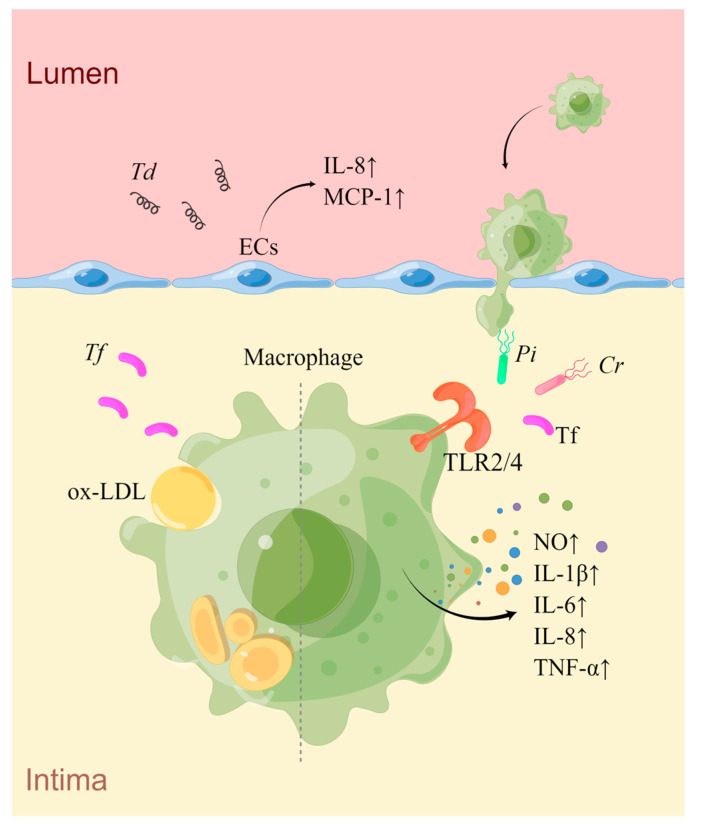
An overview of the mechanisms of *Pi*, *Tf*, *Td*, and *Cr* in AS. *Td* infection induces the secretion of IL-8 and MCP1 by ECs, promoting the adhesion and aggregation of monocytes towards the sub-endothelial space. *Pi*, *Cr*, and *Tf* can activate macrophages via the TLR2/4 signaling pathway, leading to increased production of inflammatory mediators. Additionally, *Tf* can also promote macrophage-derived foam cell formation induced by ox-LDL (drawn by Figdraw).

**Table 1 ijms-24-12861-t001:** The roles of *Pg* in AS-related cells.

Targeting Cell	Effection on Cells	Pathways	Association with AS of Mice	Association with AS of Human
ECs	Upregulation of adhesion factors and chemokine	TLRs signaling pathway [70,71,72]; NF-κB pathway [83,87,88]; MIF-CD47-CXCR4 receptor–ligand complex [90]	Plaque growth [59,60,61,62]; increased serum IgG against *Pg* and systemic pro-inflammatory cytokines [60,65]; plaque instability [58,59]; cardiac rupture [66]	High detective rate in AS plaques [45,51,52,53]; high serum IgG to *Pg* and IgG to *Pg* HSP60 [55]; more serious carotid stenosis for type 2 DB with more serum anti-*Pg* IgG [56]; patients; abdominal aortic aneurysm [57]; stroke [58]
	*Pg* intracellular survival and immune escape	Autophagy [73,74]		
	TNF-mediated cell death	Kgp initiates proteolysis of RIPK1, RIPK2, and PARP in ECs [75]		
	Suppression of cell proliferation	[76]		
	Cell anoikis	Neural cadherin and vascular endothelial cadherin cleavage and integrin β1 degradation by gingipains [77,78]		
	EndMT	Inflammation-induced by *Pg* [65]		
	Oxidative stress and inflammatory response	TLRs-NF-κB-Bmal1- NF-κB pathway [46]; NOS/BH4/Nrf2/GSK-3β pathway [81]; mitochondrial fission and dysfunction via phosphorylation and recruitment of Drp1 [82]		
Macrophages	Adhesion and aggregation to the endothelium	Through CCR2 and integrin αMβ2 [83,84,85,86,87]		
	Release of inflammation molecules	[65,91,92]		
	Lipid accumulation	modification of LDL [93] Upregulation of CD36 via the c-Jun-AP-1 pathway [94] or ERK/NF-κB [95] Upregulation of LIMP2 [96,97] and FABP4 [98] Degradation of ABCA1 [94]; oxidation of HDL [99]		
	MMP9 activation	[109]		
VSMCs	Proliferation	TGF-β and Notch signaling pathways and so on [103]; upregulation of the Angpt2/Angpt1 ratio [107]		
	Differentiation and calcification	Upregulation of Runx2 via ERK1/2 signaling by OMV of *Pg* [104]		
	Phenotype transition	Upregulation of the Angpt2/Angpt1 ratio [100,106,107]		
Th17 and Treg cells	Th17/Treg imbalance	Upregulation of IL-6 [63]		
Platelets	Aggregation	Interaction between Hgp44 of *Pg*, reactive IgG, Fc γRIIa receptors, and GPIb α receptors on platelet surface [111]		

**Table 2 ijms-24-12861-t002:** The roles of *Aa* in AS-related cells.

Targeting Cell	Effection on Cells	Pathways	Association with AS of Mice	Association with AS of Human
ECs	Upregulation of ICAM-1 and VCAM-1;	[135]	Plaque lipid deposition [128]; proatherogenic lipoprotein profiles [129]; increased serum ox-LDL and oxidative stress [128]; elevated serum inflammatory factors and chemokines [130], upregulation of innate immune signaling molecules, adhesion molecules, chemokines, LOX-1, and hsCRP in the aorta [130]	A 55.5% positivity in plaques among elderly AS patients [118]; higher coronary artery calcification, thicker carotid IMT, and lower ABI [119]; ruptured cerebral aneurysms [120]; stable CAD [121]; ACS [121]
	suppression of cell proliferation; promotion of cell apoptosis	LtxA arrests G2/M phase of cell cycle [135]		
Monocytes	Pyroptosis, release of inflammatory cytokines	LtxA of *Aa* binds with LFA-1 and activates the inflammasome [134]		
	aggregation and adhesion	Upregulation of ICAM-1 [136,137]		
	Release of inflammatory cytokines, increased cholesterol accumulation	Downregulation of SR-BI and ABCA1 [138]		

**Table 3 ijms-24-12861-t003:** The roles of *Fn* in AS-related cells.

Targeting Cell	Effection on Cells	Pathways	Association with AS of Mice	Association with AS of Human
ECs	Increased permeability Impaired proliferation and induced apoptosis	Reduction of EC adhesion molecule-1 [153] [154,155]	Increased plasma TG and cholesterol levels [150,151]; elevated serum levels of IL-6, CRP, and LDL, decreased levels of HDL [152]	A positivity rate of 34% in carotid endarterectomy specimens [147]; ruptured cerebral aneurysms [120]; stable CAD [149]
	Upregulated chemotactic factors and cell adhesion molecules	[152]		
Hepatic cells	Glycolysis and lipid synthesis	PI3K/Akt/mTOR signaling pathway [150]		
Monocytes	Inflammatory responses	PI3K-AKT/MAPK/NF-κB signaling pathway [156]		
	M1 polarization; cell apoptosis cholesterol accumulation	modulation of lipid metabolism-related genes including *SR-A1*, *ACAT-1*, *ABCA1* and *ABCG1* [151]; upregulation of FABP4 via a JNK-dependent mechanism [98]		

**Table 4 ijms-24-12861-t004:** The roles of *Pi*, *Tf*, *Td*, and *Cr* in AS-related cells.

Bacteria	Targeting Cell	Effection on Cells	Pathways	Association with AS of Mice	Association with AS of Human
*Pi*	Macrophages	Release of inflammatory cytokines	TLR4 signaling pathway [174];	-	Stable CAD and ACS [169]; thicker intimamedia [170]; CHD in smokers [171]; stroke [172]
		IL-8 production	CD14 and TLR2 signaling pathway activated by PCG of Pi [176]		
*Tf*	Macrophages	Release of pro-inflammatory mediators	[178,179,180]	Plaque enlargement, increased serum levels of CRP and LDL, decreased serum level of HDL, and expression of cholesterol efflux-related gene expression in liver [181]; lowered serum NO level and increased SAA [182]; intraplaque hemorrhage [183]	CAD and ACS [169]
		Foam cell formation	[181]		
*Td*	ECs	Facilitating chemotaxis and aggregation of monocytes to the subendothelium	Upregulation of IL-8 and MCP-1 [184]	Enlarged arterial plaques, decreased serum NO levels, and increased serum VLDL and ox-LDL levels [185]	CHD in ever smokers [171]; elevated levels of TG and reduced levels of HDL [186]
*Cr*	Macrophages	IL-6 secretion	TLR4 signaling pathway [187]	-	Stable CAD and ACS [149]; thick carotid arterial walls [188]

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
