# Peer review of "The Roles of Periodontal Bacteria in Atherosclerosis"

_ijms, 2023, doi:10.3390/ijms241612861_

Round 1

Reviewer 1 Report

Identification of periodontal bacteria from human samples has been described by Sanz M et al (Sanz et al. Methods of detection of Actinobacillus actinomycetemcomitans, Porphyromonas gingivalis and Tannerella forsythensis in periodontal microbiology, with special emphasis on advanced molecular techniques: A review. J Clin Periodontol. 2004), and should be cited in this review.

A formal identification of P. gingivalis in atherothombotic plaques was also shown by Brun et al. (Brun A.,  Innovative application of nested PCR for detection of Porphyromonas gingivalis in human highly calcified atherothrombotic plaques, J Oral Microbiol 2020).

According to Papapanou et al. the term “Chronic Periodontitis” is now archaic. Please change this term to those adapted from the current classification (Papapanou P et al. Periodontitis: Consensus report of workgroup 2 of the 2017 World Workshop on the Classification of Periodontal and Peri-Implant Diseases and Conditions. J Periodonto 2018)

Reviewer 2 Report

Interesting subject. The reading has a good flow. The paper is more like a narrative, textbook-like review - an overview of the periodontal pathogens that may have a role in atherogenesis. The English language is good. The aim is precise. The explanations are extensive and sometimes difficult to read/follow but the research on the field is impressive. It may attract readers with an interest in the study of atherosclerosis. Some specific comments:

- the phrase "microbial infections such as herpes simplex virus, Chlamydia pneumoniae, Porphyromonas gingivalis (Pg), Helicobacter pylori, influenza A virus, hepatitis C virus, cytomegalovirus (CMV), and HIV have all been identified as risk factors for AS." needs a reference. 

- the review has no figure. Is it possible maybe to draw one central figure that summarizes/illustrates all 8 pathogens studies by the authors? (as the reading is often times heavy / mechanistic).

- briefly discuss (either in Introduction or Discussion) about the topographical aspect of the association between periodontal disease and atherosclerosis. For example, is there evidence that a specific germ contributes more for peripheral artery disease or coronary disease? There is robust evidence for a positive association between periodontitis and coronary heart disease and stroke. A very recent consensus on this topic has just been published, you may wish to cite it (https://doi.org/10.1111/jcpe.13807). Furthermore, say a few words about periodontal disease and PAD (cite DOI: 10.3390/ijerph19169801); PAD patients had more missing teeth than control subjects did (doi: 10.1186/s12872-018-0879-0).

- Change "10. Summary" to Conclusions

As stated before, the English language is very good. 

Reviewer 3 Report

This is a nicely composed literature review.

The authors could have made this into a scoping/ systematic review which would provide more stronger evidence if available about the discussed process of periodontal pathogens and AS formation.

Although the authors have discussed the available literature, this does not add any new or unique piece of information than that is already available.

I would encourage the authors to utilize the available information and to make this into a scoping/ systematic review and resubmit the paper.

Other points to be note are:

1. Bacteria genus name and species names to italicized.

2. Tannerella forsythensis: is a former name and the current name for this bacteria is Tannerella forsythia.

3. Lines 133-136: Check for grammar.

Mentioned above

Reviewer 4 Report

Reviewer

Manuscript ID: ijms-2453092
Type of manuscript: Review
Title: The Roles of Periodontal Bacteria in Atherosclerosis
Authors: Xiaofei Huang, Mengru Xie, Xiaofeng Lu, Feng Mei, Wencheng Song,
Yang Liu *, Lili Chen *
Submitted to section: Molecular Immunology,

This review analyzes the complex influence of periodontitis bacteria on Atherosclerosis . The aim is to inform our understanding of the relationship between periodontal pathogens and Atherosclerosis.

It should be emphasized that the bacteria incriminated in this article are naturally harmless commensals in the presence of a healthy periodontium. They become pathogenic beyond a certain threshold. Periodontitis results from this ecological imbalance (dysbiosis) within the resident commensal microbial community of dental plaque. This phenomenon promotes the growth of initially harmless species present in small proportions and which therefore become pathogenic. Among more than 1000 commensal bacterial species present in the oral cavity, some have been strongly associated with clinical diagnoses of periodontitis and more particularly with advanced periodontal lesions.

Regarding the bacteria incriminated in the article, it should be emphasized that periodontal disease is not the only source for these microorganisms. Indeed, at the level of the digestive tract, microbial dysbiosis can also be at the origin of the diffusion of pathogens. I would therefore be less categorical in the conclusion of the article.

The American Heart Association (AHA) supports an independent association between periodontal disease (PD) and atherosclerotic vascular disease, but not a causal relationship between the two ..Lockhart PB, Bolger AF, Papapanou PN, Osinbowale O, Trevisan M, Levison ME, et al. Periodontal disease and atherosclerotic vascular disease: Does the evidence support an independent association?: a scientific statement from the American heart association. Circulation (2012) 125(20):2520–44. doi: 10.1161/CIR.0b013e31825719f3

Unfortunately, single-cell sequencing studies on P. gingivalis and AS have not been reported yet. It is believed that shortly, the use of single-cell sequencing technology will certainly provide important theoretical support for the prevention and treatment of P. gingivalis to accelerate the progression of AS.  Front. Immunol., 14 March 2023 Sec. Inflammation
Volume 14 - 2023 | https://doi.org/10.3389/fimmu.2023.1103592

The entire article must be updated at the level of the bibliography.

Abstract

Line 20.  Correction (deseases ? )

Introduction

Line 48.  Necessary of reference for this sentence.

Line 56. Necessary of reference for this sentence.

Line 91. Concerning Porphyromonas gengivalis too old reference (14) 2007. More recent publication..

 KrÄ™gielczak A, Dorocka-Bobkowska B, SÅ‚omski R, Oszkinis G, KrasiÅ„ski Z (2022) Periodontal status and the incidence of selected bacterial pathogens in periodontal pockets and vascular walls in patients with atherosclerosis and abdominal aortic aneurysms. PLoS ONE 17(8): e0270177. https://doi.org/10.1371/journal.pone.027017

 Recently  2022  Kregielczak A et al  detect  by polymerase chain reaction (PCR), periodontitis pathogens: Porphyromonas gingivalis, Tanarella forsythia, Aggregatibacter actinomycetem- comitans, Prevotella intermedia and Treponema denticola have been demonstrated in patients with Severe chronic generalized periodontitis in stages III and IV. Among the periodontal pathogens examined, only Porphyro monas gingivalis were confirmed in 1 sample of atheromatous plaque taken from the wall of an aortic aneurysm. Therefore, the presence of this bacterium in these vessels was considerate occasionally in patients with chronic periodontitis.

Line 266.  More recent ref concerning Aggregatibacter actinomycetemcomitans

Iman Razeghian-Jahromi  et al.. Prevalence of Microorganisms in Atherosclerotic Plaques of

Coronary Arteries: A Systematic Review and Meta-Analysis Volume 2022, Article ID 8678967, 12 pages

https://doi.org/10.1155/2022/8678967

Akhi R et al 2022.  highlight the  role  of  A.  actinomycetemcomitans  virulence  factors  in  early-stage  periodontitis  lesions  by  activating  the  humoral  immune  response.

Akhi, R., Nissinen, A. E., Wang, C., Kyrklund, M., Paju, S., Mäntylä, P., Buhlin, K., Sinisalo, J., Pussinen, P. J., & Hörkkö, S. Salivary IgA antibody to malondialdehyde–acetaldehyde associates with mild periodontal pocket depth. Oral Diseases. 2022;28:2285–2293. https://doi.org/10.1111/odi.13936

 Line 329. More recent reference concerning  Fusobacterium nucleatum

Zixin Fana, Pengzhou Tanga,Cheng Lia, Qi Yanga,Yan Xua, Chuan Sud and Lu Lia.

Fusobacterium nucleatum and its associated systemic diseases: epidemiologic studies and possible mechanisms JOURNAL OF ORAL MICROBIOLOGY 2023, VOL. 15, 2145729

https://doi.org/10.1080/20002297.2022.2145729

Line 371 : concerning   Prevotella intermedia  too  old references (110-118). Please actualize the ref .

For examples…

Schenkein HA et al.  Mechanisms underlying the association between periodontitis and atherosclerotic disease. Periodontol 2000. 2020 Jun;83(1):90-106. doi: 10.1111/prd.12304. PMID: 32385879

Boukobza M et al. Premier cas d'endocardite infectieuse sur valve aortique prosthétique dû à Prevotella Intermedia.

 Ann Cardiol Angeiol (Paris). 2022 Oct;71(4):240-242. doi: 10.1016/j.ancard.2022.04.001. Epub 2022 Aug

Round 2

Reviewer 2 Report

I have read the revision of Huang et al. I was glad to see a point-by-point personalized response and that the article significantly improved. With the new figures, it looks much better. Reading again carefully their review, I have delved further more into this topic and I have some final to-the-point suggestions for the authors which I kindly ask them to add before publication:

They are mostly about my previous comment about Coronary/Carotid/PAD and specific periodontal disease associations (which you already began to address and added consistent new information):

1.       Make a smoother transition between the phrase with coronaries and the one with carotids. After the phrase “Periodontitis has been demonstrated to correlate with an elevated risk of CHD (OR = 1.24 (95% CI 1.01-1.51) to 1.34 (95% CI 1.10-1.63)), as well as an increased likelihood of experiencing a myocardial infarction (OR = 1.49; 95%CI = 1.21-1.83) [24].” please squeeze in the following phrase “Similar to coronary arteries, in carotid atherosclerosis, it is crucial to consider not only the extent of stenosis but also certain distinctive features of vulnerable plaques. These characteristics encompass intraplaque hemorrhage, plaque ulcerations, relative volume of calcification, and overall plaque burden. These factors have been demonstrated to offer distinct supplementary insights into the likelihood of recurrent stroke and in-stent restenosis” and cite the following 2 references: doi: 10.1016/j.jcmg.2022.04.003 & https://doi.org/10.1155/2022/4196195

2.       To the phrase “Peripheral arterial disease (PAD) is a prevalent condition affecting the blood vessels of the limbs. It manifests as insufficient blood flow to the limbs due to arterial stenosis, commonly resulting in intermittent claudication and other associated symptoms [28]” please add the following additional reference: https://doi.org/10.3390/ijerph19169801

3.       Add (wherever you may consider appropriate) the concept of in-stent restenosis associated with periodontal disease, a new possible causation relationship (and add these 2 references: https://doi.org/10.3389/fdmed.2021.673626 & https://doi.org/10.3390/jpm13050760)

I congratulate the authors for their comprehensive work. 

No major English language issues. 

Reviewer 3 Report

Thank you for the response.

With the current changes and introduction of the oral gut microbiota relationship that has been made there is a visible break in the flow of the paper. 

The Lines 595 and 599 needs relevant references, those can also be checked in the file included.

Lines 613- 615 and the lines 635- 642, are very contradictory and will surely confuse the readers. 

Finally if the Oral, Gut Microbial axis and their relation with AS was been explored then the detailed description of some of the earlier mentioned micro-organism is not needed. The majority of the studies related to Oral, Gut and AS mention different/ larger set of organisms than those mentioned above.

If the aim is to explore and report/ review on the literature related to Oral, Gut and AS relationship then the manuscript needs a major re-write up and submitted as a different paper.

Reviewer 4 Report

Lines 304-315 : caption to figure 1 requires references for the reader.

there are many abbreviations in tables and figures that need to be spelled out for better understanding. a glossary at the end would be welcome.
